Allantoin ameliorates chemically-induced pancreatic β-cell damage through activation of the imidazoline I3 receptors

Amitani Marie 1
Cheng Kai-Chun 1
Asakawa Akihiro 1
Amitani Haruka 1
Kairupan Timothy Sean 1
Sameshima Nanami 1
Shimizu Toshiaki 2
Hashiguchi Teruto 2
Inui Akio 1 inui@m.kufm.kagoshima-u.ac.jp
1 Department of Psychosomatic Internal Medicine, Kagoshima University Graduate School of Medical and Dental Sciences , Kagoshima , Japan
2 Department of Laboratory and Vascular Medicine, Kagoshima University Graduate School of Medical and Dental Sciences , Kagoshima , Japan
Marunaka Yoshinori
Electronic publication date: 2015 Aug 6
Publication date: 2015
Volume: 3
Electronic Location ID: e1105
Received 2015 Apr 24; Accepted 2015 Jun 25
Copyright: © 2015 Amitani et al.
Copyright year: 2015
Copyright holder: Amitani et al.
License: This is an open access article distributed under the terms of the Creative Commons Attribution License, which permits unrestricted use, distribution, reproduction and adaptation in any medium and for any purpose provided that it is properly attributed. For attribution, the original author(s), title, publication source (PeerJ) and either DOI or URL of the article must be cited.
License URL: https://creativecommons.org/licenses/by/4.0/

Keywords: Allantoin, Imidazoline 3 receptor, Pancreatic, PLC-related pathway, Streptozotocin

Funding: The authors declare there was no funding for this work.

==============================
Objective. Allantoin is the primary active compound in yams (Dioscorea spp.). Recently, allantoin has been demonstrated to activate imidazoline 3 (I3) receptors located in pancreatic tissues. Thus, the present study aimed to investigate the role of allantoin in the effect to improve damage induced in pancreatic β-cells by streptozotocin (STZ) via the I3 receptors.

Research Design and Methods. The effect of allantoin on STZ-induced apoptosis in pancreatic β-cells was examined using the ApoTox-Glo triplex assay, live/dead cell double staining assay, flow cytometric analysis, and Western blottings. The potential mechanism was investigated using KU14R: an I3 receptor antagonist, and U73122: a phospholipase C (PLC) inhibitor. The effects of allantoin on serum glucose and insulin secretion were measured in STZ-treated rats.

Results. Allantoin attenuated apoptosis and cytotoxicity and increased the viability of STZ-induced β-cells in a dose-dependent manner; this effect was suppressed by KU14R and U73112. Allantoin decreased the level of caspase-3 and increased the level of phosphorylated B-cell lymphoma 2 (Bcl-2) expression detected by Western blotting. The improvement in β-cells viability was confirmed using flow cytometry analysis. Daily injection of allantoin for 8 days in STZ-treated rats significantly lowered plasma glucose and increased plasma insulin levels. This action was inhibited by treatment with KU14R.

Conclusion. Allantoin ameliorates the damage of β-cells induced by STZ. The blockade by pharmacological inhibitors indicated that allantoin can activate the I3 receptors through a PLC-related pathway to decrease this damage. Therefore, allantoin and related analogs may be effective in the therapy for β-cell damage.

Introduction

Allantoin is the primary active compound in yams (Dioscorea spp.) (Sagara et al., 1989). In the pharmaceutical industry, yams are widely used to prevent inflammation and ulcers because they contain ureides such as allantoin (Lee et al., 2010). Dioscoreaceae plants have also been shown to improve metabolic and diabetic disorders (Chang et al., 2005; Sato et al., 2009; Wang et al., 2012).

Several pathogenic processes are involved in the development of diabetes, including the destruction of pancreatic β-cells that results in insulin resistance (Cnop et al., 2005). Autoimmunity is one of the main causes of diabetes type 1 via damage of the insulin-producing β-cells in the pancreas (American Diabetes, 2010). In addition to insulin resistance, increased apoptosis and a significant reduction in the number of β-cells have been implicated in type 2 diabetes (Butler et al., 2003). Thus, prevention of pancreatic damage and the development of therapeutic strategies to protect β-cells have been introduced as a major target for the management of diabetes (Mandrup-Poulsen, 2001).

The imidazoline receptor is an orphan receptor with three subtypes. The imidazoline 1 (I1) receptors act to lower blood pressure (Ernsberger et al., 1995), whereas the imidazoline 2 (I2) receptors serve as an allosteric binding site for monoamine oxidase and are known to be involved in pain modulation, neuroprotection and increased glucose uptake in muscle cells (Li & Zhang, 2011; Lui et al., 2010). The imidazoline 3 (I3) receptors play an important role in regulating insulin secretion from β-cells in the pancreas (Head & Mayorov, 2006).

Guanidine derivates can bind to the imidazoline receptors (Dardonville & Rozas, 2004). Allantoin, a guanidine derivative, has been shown to activate the I1 receptors to attenuate hyperlipidemia, improve hepatic steatosis and act as an antihypertensive agent (Chen et al., 2014a; Yang et al., 2012). Additionally, allantoin also increases glucose uptake in muscle cells via the I2 receptors (Chen et al., 2012; Lin et al., 2012). A recent study demonstrated that allantoin was able to bind to the I3 receptors, resulting in the lowering of blood glucose due to increased plasma insulin levels (Tsai et al., 2014). Moreover, insulinotropic agents such as glucagon-like peptide-1, an incretin derived from the transcription product of the proglucagon gene, can also protect β-cells from apoptosis (Cernea, 2011; Liu et al., 2012). Thus, we speculated that allantoin may play a role in pancreatic β-cell protection via the I3 receptor. The present study aimed to identify the role of allantoin in improving damage in pancreatic β-cells induced by a low dose of streptozotocin (STZ).

Material and Methods

Animals

Male Wistar rats weighing 320–340 g obtained from Japan SLC, Inc (Shizuoka, Japan), were maintained in an environment under a 12 h light/12 h dark cycle with a controlled room temperature at the animal center of Kagoshima University (Kagoshima, Japan). Food and tap water were provided ad libitum with free access. All procedures in this study were approved by the Ethics Committee for Animal Care and Use of Kagoshima University (IRB approval number MD14059).

Islet isolation and primary culture

Following a previous method (Shewade, Umrani & Bhonde, 1999), pancreatic islets were removed from rat pancreases. The pancreases were incised into smaller portions and digested with 1 mg/mL collagenase (Roche, Basel, Switzerland) for 10 min. The digested samples were washed two times with RPMI 1640 medium (Sigma, St. Louis, Missouri, USA) containing 10% fetal bovine serum FBS (Thermo, Waltham, Massachusetts, USA) to inactivate the collagenase. The isolated pancreatic islets were cultured in RPMI 1640 supplemented with 1% penicillin and streptomycin (Life Technology, Carlsbad, California, USA), 1% amphotericin B (Sigma, St. Louis, Missouri, USA) and 10% FBS. The primary culture was incubated (37 °C with 5% CO2) for 48 h. After the incubation period, the primary culture was divided into 6-well plates for Live/Dead double staining assay, 96-well plates for the ApoTox-Glo Triplex assay, and 12-well plates for annexin and flow cytometry analysis.

Treatment of cells with reagents

The primary cultured cells were devided into 6-well plates. The medium was removed, and the cells were washed once with phosphate-buffered saline (PBS). RPMI 1640 medium containing 25 mM glucose was added to each well with 5 mM STZ (Sigma-Aldrich, St. Louis, Missouri, USA) and incubated for 6 h to induce cell apoptosis. To know the role of allantoin in the protection of pancreatic β-cells against STZ, allantoin (Sigma-Aldrich, St. Louis, Missouri, USA) pretreatment at various doses was provided before 30 min prior to the addition of 5 mM STZ and incubated for 6 h. To identify the signaling pathway of allantoin in β-cells, 1 µM KU14R (Santa Cruz Biotechnology, Santa Cruz, California, USA): an I3 binding site antagonist, or 1 µM U73122 (TOCRIS, Bristol, UK): the phospholipase C (PLC) inhibitor were provided before 30 min prior to the addition of allantoin as previously described before (Yang et al., 2015). All the medium was removed, and the cells were washed three times with PBS prior to processing for the evaluation of morphology.

Live/dead double staining assay

Using the Live/dead assay kit (Life Technology, Carlsbad, California, USA), we stained β-cells to distinguish the living cells from dead cells according to the manufacturer’s instruction. We added 100 µl of Live/Dead solution to the samples and incubated them for 15 min at room temperature. Then, the staining solution was removed and the samples were viewed under a fluorescence microscope (LSM700)(Zeiss, Jena, Germany). Living cells were detected at green fluorescence, whereas dead cells were detected at red fluorescence.

ApoTox-Glo triplex assay

The β-cells were seeded into 96-well plates at a total density of 1 × 104 cells per well. Each well contained 200 µl RPMI 1640 medium and the test compound where appropriate. ApoTox-Glo Triplex Assay (Promega, Madison, Wisconsin, USA) was used according to the manufacturer’s instructions to measure the β-cells’ viability, cytotoxicity, and apoptosis. After 24 h the Viability/Cytotoxicity reagent, containing both the GF-AFC substrate and the bis-AAF-R110 substrate, was added to all wells and incubated for 30 min. Caspase-Glo 3/7 was added to the wells and mixed briefly for 30 s, then incubated for 30 min at room temperature. Fluorescence was measured at 380EX/510EM to assess viability, 485EX/520EM to assess cytotoxicity, and luminescence was mesured to assess apoptosis.

Annexin V/PI staining and flow cytometry analysis

The primary cultured β-cells were divided into 12-well plates and categorized into four groups. Each group was treated with different reagents as follows: (1) 5 mM STZ; (2) 5 mM STZ and 100 µM allantoin; (3) 5 mM STZ, 1 µM KU14R and 100 µM allantoin; and (4) control. The cells were incubated with the reagents for 48 h. Then, the cells were collected and the apoptotic cells in each group were quantified using Annexin V-PI staining (Life Technology, Carlsbad, California, USA) and analyzed using flow cytometer based on the previously described method (Luo et al., 2014).

Western blotting analysis

Western blotting analysis was performed to determine caspase-3 and Bcl-2 expression. The β-cells were pre-cultured with 5 mM STZ for 6 h prior to the addition of 100 µM allantoin with or without 1 µM KU14R or vehicle for 30 min. The β-cells were washed with ice-cold PBS and incubated for 15 min to allow lysis to occur. The protein concentration was measured by BCA protein assay (Thermo Fisher Scientific Inc., Waltham, Massachusetts, USA). The protein samples were filtered and separated by SDS-PAGE (Polyacrylamide Gel Electrophoresis) (10% acrylamide gel) using the Bio-Rad Trans-Blot system and were transferred to polyvinylidene difluoride membranes. The membrane was blocked with 5% non-fat milk in Tris-buffered saline containing 0.1% Tween 20 (TBS-T). The membran was incubated for 2 h and washed with TBS-T and hybridized overnight with primary antibodies, caspase-3 (Merk Millpore, USA) and Bcl-2 (Cell signaling Technology, Danvers, Massachusetts, USA), diluted with a suitable concentration of TBS. Incubation with secondary antibodies and the detection of the antigen-antibody complex was performed using an ECL kit (Thermo Fisher Scientific Inc., Waltham, Massachusetts, USA). The bands densities were quantified using a laser densitometer.

Glucose and insulin levels in STZ-treated rats

The induction of pancreatic cell damage was accomplished by injecting 45 mg/kg STZ dissolved in 10 mM Na-citrated buffer intraperitoneally. STZ-treated rats with blood glucose above 200 mg/dl at 7 days post-injection were included in the group. Total of 24 rats were divided into three groups as follows: Control (STZ) (n = 8), STZ + allantoin (n = 8), STZ + KU14R + allantoin (n = 8). The third group was treated with an intravenous injection of 8 mg/kg/day KU14R; the first and second groups were treated with the same volume of vehicle injected intravenously. After 30 min of KU14R injection, the second and third groups received 10 mg/kg/day of allantoin intravenously. The first group was injected the same volume of vehicle intravenously. The experiments were performed for 8 days. The blood samples were obtained from tail vein everyday. The plasma glucose levels were measured everyday, and the plasma insulin levels were measured on day 0, 4, 6, 8.

Statistical analysis

Statistical analyses were performed using SPSS software (SPSS, Inc., Chicago, Illinois, USA). An analysis of variance (ANOVA) with Tukey’s test to determine significant differences were used to compare multiple treatment groups. Data are presented as the mean ± standard error (S.E.) based on the number (n) of samples in each group. Statistical significance was set at p < 0.05.

Results

Allantoin decreased streptozocin-induced cytotoxicity and apoptosis in β-cells

Viable β-cells were significantly reduced in the STZ treated group, while cell toxicity and apoptosis were significantly increased compared to the control. In contrast, treatment with allantoin at various concentrations (1 µM, 10 µM, and 100 µM) significantly increased cell viability, and decreased cytotoxicity and apoptosis induced by STZ in a dose-dependent manner. These results suggest that allantoin attenuated STZ-induced cell damage (Fig. 1).

Figure 1 ApoTox-Glo triplex assay showing the viability (A), cytotoxicity (B), and apoptosis (C) of β-cells treated with 5 mM streptozotocin (STZ), 5 mM STZ + 1 µM allantoin, 5 mM STZ + 10 µM allantoin, and 5 mM STZ + 100 µM allantoin (n = 6 for each group).

Data are presented as the mean ± SE. ∗P < 0.05, ∗∗P < 0.01.

Allantoin-induced increase in β-cell viability was blocked by an I3 antagonist

I3 receptors located on pancreatic β-cells are known to stimulate insulin secretion (Chen et al., 2014b; Dardonville & Rozas, 2004). Thus, the effect of allantoin on imidazoline I3 receptors was investigated using KU14R, an I3 specific antagonist, in the Apo Tox triplex assay. The effect of allantoin was inhibited by KU14R (Fig. 2).

Figure 2 ApoTox-Glo Triplex assay showing the viability (A), cytotoxicity (B), and apoptosis (C) of β-cells in rats treated with 5 mM streptozotocin (STZ), 5 mM STZ + 100 µM allantoin, 5 mM STZ + 1 µM KU14R + 100 µM allantoin (n = 6 for each group).

Data are presented as the mean ± SE. ∗∗P < 0.01.

Allantoin increased the viability of STZ-treated β-cells

Exposure of the β-cells to 5 mM STZ induced apoptosis based on the increase in red fluorescence compared to the control group. The β-cell viability was increased by the addition of allantoin, shown by the marked decrease in red fluorescence emitted by EthD-1, which can bind to the DNA of the dead cells. We found that pretreatment with 1 µM KU14R for 30 min resulted in the increased EthD-1 binding, thereby reducing the action of allantoin (Fig. 3).

Figure 3 Cell viability image of β-cell after treatment with 5 mM streptozotocin (STZ), 5 mM STZ + 100 µM allantoin, 5 mM STZ + 1 µM KU14R + 100 µM allantoin.

Allantoin greatly improved the viability of STZ-induced β-cells apoptosis.

Allantoin decreased the β-cell apoptosis percentage detected by flow cytometry

Viable cells are located in the lower left quadrant in the flow cytometric analysis. Treatment with 5 mM STZ for 6 h increased the percentage of apoptotic cells to 45.5%, indicated by the increased number of cells in the lower right quadrant. Allantoin reversed this effect and decreased the percentage of apoptotic cells to 32.7%, as shown by movement of the cell population from the lower right quadrant to the lower left quadrant. Co-treatment with KU14R blocked the action of allantoin and induce apoptosis to 53.9% (Fig. 4).

Figure 4 Flow cytometry of apoptotic cells.

Samples were incubated with FITC-labeled Annexin-V and propidium iodide. Number at the corner represent the percentage of cells found in each quadrant.

Allantoin-induced cell protective effect involved phospholipase C

To test whether the protective effect of allantoin involves the PLC pathway, we applied U73122: a PLC inhibitor. U73122 attenuated the protective effect of allantoin in β-cells (Fig. 5).

Figure 5 ApoTox-Glo Triplex assay showing the viability (A), cytotoxicity (B), and apoptosis (C) of β-cells in rats treated with 5 mM streptozotocin (STZ), 5 mM STZ + 100 µM allantoin, 5 mM STZ + 1 µM U73122 + 100 µM allantoin (n = 6 for each group).

Data are presented as the mean ± SE. ∗∗P < 0.01.

Allantoin decreased caspase-3 and increased Bcl-2 expression

Caspase-3 and Bcl-2 are associated with the process of cell death. Caspase-3 is known to play a central role in cell apoptosis, whereas Bcl-2 regulates cell death. We used Western blotting analysis to detect the expression of these two regulatory proteins. In the STZ-treated group, the expression level of caspase-3 was significantly increased, while the Bcl-2 level was significantly decreased. Allantoin significantly suppressed the expression of caspase-3 and significantly increased the expression of Bcl-2. In contrast, KU14R significantly inhibited these actions of allantoin (Fig. 6).

Figure 6 Western blotting analysis of the expression levels of caspase-3 and Bcl-2.

The expression level of caspase-3 was reduced by allantoin (A), while Bcl2 expression was increased (B) (n = 6 for each group). Data are presented as the mean ± SE. ∗P < 0.05, ∗∗P < 0.01.

Plasma glucose levels in rats treated with STZ, allantoin, and KU14R

Plasma glucose levels were increased following intraperitoneal injection of STZ. Treatment with allantoin significantly lowered the blood glucose levels in STZ-treated rats. This effect of allantoin was countered by combined treatment with KU14R (Fig. 7).

Figure 7 Effects of allantoin and KU14R on blood glucose levels in streptozotocin (STZ) -treated rats.

STZ-treated rats were daily treated with 10 mg/kg allantoin and 8 mg/kg KU14R. Plasma glucose levels were measured daily for 8 days (n = 8 for each group). Values are presented as the mean ± SE. ∗P < 0.05 and ∗∗P < 0.01 for the difference between STZ and STZ + Allantoin + KU14R. # < 0.05 for the difference between STZ + Allatoin and STZ + Allantoin + KU14R.

Plasma insulin levels in rats treated with STZ, allantoin, and KU14R

Plasma insulin levels were decreased following intraperitoneal injection of STZ. Treatment with allantoin significantly improved the plasma insulin levels. In contrast, co-treatment with KU14R countered this effect of allantoin (Fig. 8).

Figure 8 Effects of allantoin and KU14R on plasma insulin levels in streptozotocin (STZ) -treated rats.

STZ-treated rats were daily treated with 10 mg/kg allantoin and 8 mg/kg KU14R. Plasma insulin levels were measured on day 0, 4, 6, 8 (n = 8 for each group). Values are presented as the mean ± SE. ∗∗P < 0.01 for the difference between STZ and STZ + Allantoin or STZ + Allantoin + KU14R. # < 0.05 and ## < 0.01 for the difference between STZ + Allatoin and STZ + Allantoin + KU14R.

Discussion

Allantoin is known to bind to the imidazoline receptors (Chung, Lee & Cheng, 2013; Tsai et al., 2014). In the present study, we found that allantoin could activate the I3 receptors to protect β-cells from the damage induced by STZ.

Diabetic disorders are associated with progressive β-cell failure and apoptosis (Eizirik & Mandrup-Poulsen, 2001). STZ induced diabetes is well-established and accepted in studies of pathogenesis as well as the complication of diabetes (Szkudelski, 2001), and it has been widely used in experimental animals (Rees & Alcolado, 2005) and cytotoxic effect of STZ in β-cells (Lenzen, 2008). To mimic this disorder, we treated β-cells with STZ (5 mM) in medium containing 25 mM glucose for 6 h. Moreover, high glucose is known to increase pancreatic cell vulnerability to toxic damage by increasing the expression of potential autoantigens on the cell membrane surface (Mellado-Gil & Aguilar-Diosdado, 2004). Therefore, we established a model that induced significant changes in β-cells, including the induction of an apoptotic response.

In the present study, we found that allantoin attenuated the damage induced by STZ in a dose-dependent manner, resulting in the reduction of STZ-induced β-cell apoptosis. Allantoin induced a significant decrease in caspase-3 expression and an increase in Bcl-2 expression detected by Western blotting. Caspases activation plays an important role in the execution phase of cell apoptosis (Jin & El-Deiry, 2005; Lee, Abouhamed & Thevenod, 2006), while Bcl-2 is considered to act as an anti-apoptotic protein that promotes cell survival (Vaux, Cory & Adams, 1988). Thus, allantoin can increase the survival rate of β-cells through improvement of apoptosis.

Imidazoline compounds have been suggested to induce insulin secretion from pancreatic β-cells through activation of the I3 receptors located on the β-cells (Tsai et al., 2014). In the presence of KU14R, I3 receptor antagonist, the protective effect induced by allantoin in β-cells was partially blocked. Flow cytometric analysis also supported this findings. As shown in Fig. 4, cell viabilities were improved after allantoin treatment, and this effect was also suppressed by the blockade of I3 receptors using KU14R. As previously described, rats injected with a low dose of STZ exhibited higher blood glucose and lower plasma insulin levels (Tsai et al., 2014). Allantoin improved the damaged function of β-cells in this animal model, which resulted in an increase of plasma insulin levels and a reduction of plasma glucose levels. This in vivo action of allantoin was also inhibited by KU14R to block the I3 receptors. Thus, the action of allantoin via the activation of I3 receptors was shown both in vivo and in vitro. Similar result was also observed in the action of canavanine (Yang et al., 2015).

Moreover, we found that the protective effect of allantoin was linked to the phospholipase C (PLC) pathway. In the presence of U73122, a well-known PLC inhibitor, the protective effect of allantoin was markedly reduced (Fig. 5). In theory, upon activation, PLC cleaves phosphatidylinositol 4,5-biphosphate into diacylglycerol and inositol 1,4,5-triphosphate, which may potentiate insulin secretion (Kahn et al., 2005). Whether this action is related to the protection of β-cells shall be investigated in future studies.

Additionally, several yam species (Dioscorea spp.) are also known to contain saponin and the aglycone portion of saponin called sapogenin, that are also proven to be beneficial in STZ induced diabetic rats (Omoruyi, 2008; Pessoa et al., 2015). Taken together, allantoin or yam (Dioscorea spp.) seems beneficial to treat and/or prevent the diabetes in the future.

Nevertheless, for the first time, we characterized the improvement of STZ-induced β-cell damage by allantoin via the I3 receptors both in vivo and in vitro.

Conclusion

Allantoin has the ability to increase β-cells viability and ameliorate β-cell damage through activation of the I3 receptors. Thus, allantoin and related analogs supplied as nutrients may be useful for the improvement of early stage of β-cell damage.

Supplemental Information

Supplemental Information 1 Allantoin Raw Data

Click here for additional data file.

We thank all the staff members of the Institute of Laboratory Animal Sciences, Kagoshima University (Frontier Science Research Center) who kept the animals in good condition.

Additional Information and Declarations

Competing Interests

Author Contributions

Animal Ethics

The authors declare there are no competing interests.

Marie Amitani conceived and designed the experiments, performed the experiments, analyzed the data, wrote the paper, prepared figures and/or tables.

Kai-Chun Cheng conceived and designed the experiments, performed the experiments, analyzed the data, contributed reagents/materials/analysis tools, prepared figures and/or tables.

Akihiro Asakawa conceived and designed the experiments, wrote the paper, reviewed drafts of the paper.

Haruka Amitani performed the experiments, analyzed the data.

Timothy Sean Kairupan wrote the paper, prepared figures and/or tables.

Nanami Sameshima performed the experiments, analyzed the data, prepared figures and/or tables.

Toshiaki Shimizu contributed reagents/materials/analysis tools.

Teruto Hashiguchi reviewed drafts of the paper.

Akio Inui conceived and designed the experiments, reviewed drafts of the paper.

The following information was supplied relating to ethical approvals (i.e., approving body and any reference numbers):

The Ethics Committee for Animal Care and Use of Kagoshima University.

IRB approval number MD14059.

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
