# Peer review of "Allantoin ameliorates chemically-induced pancreatic β-cell damage through activation of the imidazoline I3 receptors"

_PeerJ, doi:10.7717/peerj.1105_

## Round 0.1 · original submission · Minor Revisions

I suggest you to revise your manuscript according to the comments provided by Reviewers 1 and 3. Reviewer 2 has any concerns on your manuscript. Therefore, I am happy if you could revise your manuscript according to the comments provided by Reviewers 1 and 3.

Reviewer 1 ·

Basic reporting

This submission investigated the effect of allantoin on pancreatic damage induced by streptozotocin (STZ). Authors found that allantoin may activate imidazoline I-3 receptors to lower the damage of STZ both in pancreatic cells and animals. Thus, allantoin is useful to apply in advance.

Experimental design

This submission was designed in a good way. Authors employed the primary culture of pancreatic cells and rats to support their hypothesis. Overall, the experimental data are enough to support the conclusion.

Validity of the findings

1. Pharmacological inhibitor(s) including KU14R and U73122 used in the experiments need to show the effective dose or concentration according to reference(s).
2. Apoptosis was identified as main damage in pancreatic cells. Is this consistent to previous report(s)? Please add in the discussion.
3. Recovery of decreased plasma insulin by allantoin in STZ-damaged rats seems useful to support the effectiveness of allantoin in vivo.

Additional comments

This is an interesting study to develop allantoin in the improvement of pancreatic cell damage. Please correct the expressions as described in each comment.

Reviewer 2 ·

Basic reporting

No problem.

Experimental design

Good design.

Validity of the findings

I think allantoin and its analogs to have possibilities to be usable for the protection of the β-cell .

Additional comments

I think that this article is a excellent article.

·

Basic reporting

I suggest that the authors state more quantitatively the data, which should increase the quality of MS.

Experimental design

"No Comments"

Validity of the findings

"No Comments"

Additional comments

This article reports that treatment with Allantoin, a component of yams, protects islet beta cells against STZ and counteract STZ-induced diabetes. The methods are sound and the results overall support the conclusion, which may be of relevance for the possible application for treating diabetes. However, there are several points that should be described more carefully and quantitatively.

Comments:
1. Fig.3; No quantitative evaluation is provided.
2. In Fig.7, blood glucose and insulin change in a time-dependent manner following STZ treatment. Hence, clearly write at which day of treatment the data in Fig.3 and other figures were taken, so that the date of beta cell survival can be correlated with blood glucose and insulin levels.
3. Fig.7 shows that KU14R only partially counteracts Allantoin effect. Authors should clearly write the effect is partial.
4. In Fig.7, counteracting effect of KU14R is as large as 70% at day 4 but 10-30% at day 8. Please discuss this transiency. Since the effect of KU14R is partial, is there I3-receptor independent pathway? This should be discussed.
5. It would be nice if authors can provide more information or strategy for the future application of Allantoin or intake of Yams to treat or prevent diabetes.

---

## Round 0.2 · accepted · Accept

Dear Prof. Inui,

I have received comments regarding your revised manuscript and it was felt that you revised the manuscript appropriately according to their comments. Therefore, I am pleased to accept your revised manuscript for publication in PeerJ.

Reviewer 1 ·

Basic reporting

This submission has revised according to comments. It is suitable for publication in this journal.

Experimental design

No Comments.

Validity of the findings

No Comments.

Additional comments

This submission has been improved in a good way.